# ALIN: An Active Learning Framework for Incomplete Networks

**Tung Khong**[1]     **Cuong Pham**[1]     **Cong Tran**[1]

[1]Posts & Telecommunications Institute of Technology, Hanoi, Vietnam

## Abstract

Significant progression has been made in active learning algorithms for graph networks in various tasks. However real-world applications frequently involve incomplete graphs with missing links, which pose the challenge that existing approaches might not adequately address. This paper presents an active learning approach tailored specifically for handling incomplete graphs, termed ALIN. Our algorithm employs graph neural networks (GNN) to generate node embeddings and calculates losses for both node classification and link prediction tasks. The losses are combined with appropriate weights and iteratively updating the GNN, ALIN efficiently queries nodes in batches, thereby achieving a balance between training feedbacks and resource utilization. Our empirical experiments have shown ALIN can surpass state-of-the-art baselines on Cora, Citeseer, Pubmed, and Coauthor-CS datasets.

## 1 INTRODUCTION

The concept of graphs (or networks) has become pervasive across numerous domains, such as citation graphs and social graphs. Similar to other forms of data, graph data are undergoing rapid expansion, presently attaining substantial magnitudes. Consequently, the expanding dimensions of these graphs pose formidable challenges in attempting to analyze such type of data comprehensively.

Graph embeddings, the technique that transforms a given graph into a lower-dimensional space while preserving its underlying structural attributes and other inherent characteristics, are now gaining considerable attention in research areas [Goyal and Ferrara, 2018, Wang et al., 2023]. By generating node embeddings, a spectrum of graph analytical tasks, including but not limited to node classification, node clustering, and link prediction, can be executed with heightened efficiency, optimizing both temporal and spatial considerations [Ou et al., 2016]. The semi-supervised graph embedding algorithms typically assume the training labeled data are given, which may not be always true in real practice [Song et al., 2023]. Given a labeling budget, the strategic selection of training labeled nodes to maximize eventual performance is thus of great importance. Addressing this concern, the concept of Active Learning (AL) has been introduced as a solution [Xie et al., 2022]. AL strategies offer a highly efficient mechanism for enhancing the process of data annotation by prioritizing the identification and labeling of the most informative instances. This, in turn, serves to optimize the efficiency and overall performance of machine learning models. Significantly, the domain of graph-based tasks, including many applications such as social network analysis, recommendation systems, and biological network inference, has benefited greatly from these developments [Deng, 2022, Vatter et al., 2023].

Recent AL-based approaches on graphs often assume the underlying network is fully known [Ma et al., 2022]. However, this assumption tends to be overly simplistic as the underlying network cannot be fully observed in many real-world applications of network analyses [Valente and Pumpuang, 2007, Rice et al., 2012]. While, in theory, it is conceivable to allocate additional resources towards the exhaustive exploration of the entire network, the endeavor to acquire a comprehensive network structure frequently proves to be prohibitively costly, demanding in terms of labor, or entirely unfeasible in practice [Valente and Pumpuang, 2007]. For example, network data extracted from social media platforms bear privacy concern limitations as a substantial 52.6% of Facebook users took measures to conceal their friends' connections during a demographic analysis of Facebook in New York City in June 2011.[1] Consequently, when working with graph data, one should assume a more practical case that only a part of the network structure is available in practice [Hou et al., 2022, Teji et al., 2022, Tran et al., 2021]. This

---

[1]We refer to Dey et al. [2012] for the statistics.

raises a critical challenge: How do we adapt AL methods to effectively operate on such incomplete graphs?

To tackle this pressing challenge, we introduce a new active learning framework explicitly tuned to handle incomplete networks: ALIN (**A**ctive **L**earning for **I**ncomplete **N**etworks) [2]. We propose a framework that incorporates an edge-based scoring mechanism into the AL framework. Conventionally, AL approaches in graphs have prioritized node-centric objectives, such as optimizing node classification accuracy, which is no longer sufficient and there is an inherent need to strategically select nodes that contribute to graph completeness. However, simply introducing edge scores can compromise the primary goal of node classification, leading to reduced overall accuracy. To strike a balance between enhancing graph completeness and preserving node classification accuracy, we introduce a two-phase training. In the initial epochs, we focus on link prediction as an auxiliary task. This early phase aims to establish an effective synergy between node scores and edge scores, facilitating the creation of informative edges within the incomplete graph. In the subsequent epochs, our approach seamlessly transitions towards prioritizing the core task of node classification, ensuring that the final objective is met with high accuracy. By combining the objectives of improving graph completeness and enhancing node classification accuracy, our proposed AL framework addresses the unique challenges posed by incomplete graphs. This innovative approach not only extends the applicability of AL techniques to real-world scenarios but also opens doors to more comprehensive and accurate graph-based data analysis.

In this paper, we present a comprehensive set of contributions, each addressing a distinct facet of the active learning problem in the context of incomplete graphs:

- We introduce the novel Active Learning on Incomplete Graphs (ALIN) framework that is meticulously designed to tackle the unique challenges posed by incomplete graph structures, offering a robust end-to-end solution.

- We extend the conventional node scoring approach by introducing edge scores. This innovation caters specifically to the optimization needs of incomplete graphs, allowing for more effective query node selection.

- We propose a novel joint loss function that seamlessly combines node classification and link prediction. This integration ensures that the interplay between these two critical components is optimized. Furthermore, we introduce a method to harmonize these two losses, thereby achieving superior results in the ultimate task of node classification.

- Our contributions are substantiated through an exten-

---

[2]The source code used in this paper is available online (https://github.com/manhtung001/ALIN).

sive series of experiments conducted on datasets. These experiments not only establish the superior performance of ALIN when compared to conventional active learning methods on benchmark graphs but also underscore the robustness of our approach across various datasets and with different GNN backbones.

## 2   RELATED WORK

The framework that we proposed is related to the following three research lines.

**Active Learning**. Traditional active learning algorithms operate by querying individual samples for labeling in a sequential manner. However, such an approach proves to be suboptimal when applied to deep learning models as it frequently retrains but updates little, and it is prone to overfitting [Ren et al., 2021]. Therefore, in deep active learning, the batch-mode setting, where a diverse set of instances are sampled and queried, is more often considered. In recent years, the optimal experimental design principle [Pukelsheim, 2006, Allen-Zhu et al., 2017] motivated the machine learning community to minimize the use of training resources and avoid tuning on a validation set. Combining the settings of one-shot learning and batch-mode active learning, several recent studies [Contardo et al., 2017, Wu et al., 2019] adopted a one-step batch-mode active learning setting.

Based on the query strategy, the majority of work can be divided into three categories [Aggarwal et al., 2014]: heterogeneity-based, performance-based, and representativeness-based. Heterogeneity-based [Zhang et al., 2017] labeled the instances that are most different from the current known model. Performance-based [Guo and Greiner, 2007] minimize labeled uncertainty of the remaining unlabelled instance. Representativeness-based [Li and Guo, 2013] labeled the instance that can represent the underlying distribution of training instances

**Active Learning on Graphs**. The majority of work can be divided into four categories, including: EER, Heuristics, Uncertainty, and GraphPart. EER (Expected Error Reduction) [Zhu et al., 2003, Macskassy, 2009, Gu and Han, 2012] is a criterion in active learning that selects instances with the highest expected reduction in classification error, aiming to improve model performance efficiently. Heuristics [Macskassy, 2009, Cai et al., 2017] are rule-of-thumb strategies used in active learning to guide the selection of informative data points for labeling, often based on measures like uncertainty, diversity, or disagreement among models. Uncertainty sampling [Ma et al., 2013, Cai et al., 2017, Wu et al., 2019, Ma et al., 2022] is an active learning method that selects instances for labeling based on the uncertainty of their predicted class probabilities, targeting instances where the model is least confident in its predictions. Recently, GraphPart [Ma et al., 2022] first splits the graph into dis-

| Experiment | Method | | | | Incomplete Network | Adaptive |
| --- | --- | --- | --- | --- | --- | --- |
| | EER | Heuristics | Uncertainty | GraphPart | | |
| Zhu et al. [2003] | x | | | | No | No |
| Macskassy [2009] | x | x | | | No | Yes |
| Gu and Han [2012] | x | | | | No | No |
| Ma et al. [2013] | | | x | | No | No |
| Cai et al. [2017] | | x | x | | No | Yes |
| Wu et al. [2019] | | | x | | No | Yes |
| Ma et al. [2022] | | | x | x | No | Yes |
| **ALIN** (ours) | | | x | x | Yes | Yes |

Table 1: Summary of active learning techniques for node classification on graphs. Here, the Adaptive column indicates that the active learner is updated based on the newly labeled instances.

joint partitions and then selects representative nodes within each partition to query. It is worth noting that all prior work operated under the assumption of complete graphs, which diverges from reality given the incomplete nature characteristic of most real-world graphs. In Table 1, we summarize the aforementioned AL methods for the node classification task.

**Link Prediction** is a fundamental problem that attempts to estimate the likelihood of the existence of a link between two nodes [Lü and Zhou, 2011]. This process enhances our comprehension of the connection between specific nodes and the evolution of the entire network. Link prediction has been widely applied to a variety of fields such as biology [Lei and Ruan, 2012] and social networks [Liben-Nowell and Kleinberg, 2003, Bonchi et al., 2011]. A multitude of methodologies exist for the prediction of links within networks. Heemakshi Malhi [2016] provided an extensive survey that encompasses diverse link prediction algorithms, with a particular emphasis on scrutinizing the limitations inherent in such methods. Lei and Ruan [2012] presented an excellent survey by summarizing different approaches, introducing typical applications, and outlining future challenges of link prediction algorithms. Building upon this foundation, Martínez et al. [2016] furnished a more contemporary perspective by incorporating recent methodologies and conducting a meticulous comparative analysis of similarity-based techniques. Since it is difficult to identify a method that has the best performance in all complex networks, which strongly depends on the structural properties of the network, the authors in Wu et al. [2022] categorized various link prediction strategies, including common neighbors-based, paths-based, probabilistic and statistical models based, classifier based, and network embedding based techniques.

# 3 PROBLEM FORMULATION

In this section, we describe a formal definition of the problem of active learning on an incomplete graph under iterative batch-mode settings and introduce a uniform set of nota-tions.

Let us denote an underlying network $\mathcal{G} = (\mathcal{V}, \mathcal{E})$ with $N$ nodes $v_i \in \mathcal{V}$, edges $(v_i, v_j) \in \mathcal{E}$. Each node is associated with a feature matrix $\mathbf{X} \in \mathbb{R}^{N \times F}$, where $F$ denotes the dimensionality of the feature vector for each node. Additionally, there exists a label matrix $\mathbf{Y} \in \mathbb{R}^{N \times C}$ for labeled nodes, where $C$ represents the number of node classes. Here, $\mathbf{Y}_{ij} = 1$ indicates that node $i$ has label $j$, where $y_i$ represents the label assigned to node $i$ and $y_i = c$ denotes $c$-th element within the set $\{1, 2, \ldots, C\}$. An oracle is available to label a query node along with its associated edges, within a given labeling budget $B$. We assume $y_i$ is drawn randomly from a distribution $\mathbb{P}_{y|x_i}$ supported on $\mathbf{Y}$. We denote $\eta_c(v) = \Pr[y = c|v]$ as the probability that $y = c$ given node $v$, and $\eta(v) = (\eta_1(v), \ldots, \eta_C(v))^T$.

In this study, we follow the iterative batch-mode setting [Wu et al., 2019]. In this setting, for each iteration, we deplete a predetermined resource budget to select a batch of nodes for labeling, streamlining the querying process to minimize redundant retraining. We entail segmenting a given budget $B$ into $K$ equitably sized partitions. For each iteration $k = \{0, \cdots, K\}$, an active learning algorithm $\mathcal{A}^{(k)}$ selects $b = [B/K]$ nodes for querying, which forms a set of selected nodes, denoted as $\mathcal{Q}_b^{(k)}$. The primary objective underlying this approach is to harness the informative feedback derived from the training process, while simultaneously safeguarding against excessive resource consumption. This contrasts with the fundamental AL setup, where only a solitary node is chosen at a time, potentially imposing considerable training overhead.

Since we tackle a setting where graph data is incomplete, for each iteration $k = \{0, \cdots, K\}$, we are given an incomplete graph $\tilde{\mathcal{G}}^{(k)} = (\mathcal{V}, \tilde{\mathcal{E}}^{(k)})$ and an incomplete label set $\tilde{\mathbf{Y}}^{(k)}$, where $\tilde{\mathcal{E}}^{(k)} \subset \mathcal{E}$ and $\tilde{\mathbf{Y}}^{(k)} \subset \mathbf{Y}$ are the edge set and the set of updated node labels at $k$-th iteration, respectively. At $k$-th iteration, when $\mathcal{A}^{(k)}$ queries $b$ nodes, we obtain $\tilde{\mathbf{Y}}_q^{(k)}$ and $\tilde{\mathcal{E}}_q^{(k)}$ as the sets of newly obtained node labels and edges after querying, respectively. Additionally, we

denote $\tilde{\mathbf{Y}}_u^{(k)}$ and $\tilde{\mathcal{E}}_u^{(k)}$ as the sets of updated node labels and updated edges set at the $k$-th iteration. Thus, we have $\tilde{\mathbf{Y}}_u^{(k)} = \tilde{\mathbf{Y}}^{(k)} \cup \tilde{\mathbf{Y}}_q^{(k)}$ and $\tilde{\mathcal{E}}_u^{(k)} = \tilde{\mathcal{E}}^{(k)} \cup \tilde{\mathcal{E}}_q^{(k)}$; and the budget $b$ is the maximum number of updated node labels. In this setting, we assume that the node feature matrix $\mathbf{X}$ is fully observable.

We aim to train a GNN-based classification model $\mathcal{M}^{(k)}$ by iteratively updating its parameters $\theta^{(k)}$. The GNN model $\mathcal{M}^{(k)}$ maps $(\tilde{\mathcal{E}}_u^{(k)}, \mathbf{X})$ to prediction vectors $\hat{\mathbf{Y}}^{(k)}$ and $\hat{\mathcal{E}}^{(k)}$. From the prediction and the observation, we compute node classification loss $l_{NC}(\tilde{\mathbf{Y}}_u^{(k)}, \hat{\mathbf{Y}}^{(k)})$ and link prediction loss $l_{LP}(\tilde{\mathcal{E}}_u^{(k)}, \hat{\mathcal{E}}^{(k)})$. To combine both losses, we sum $l_{NC}$ and $l_{LP}$ with a hyperparameter weight $\beta$, denoted as $\mathcal{L}^{(k)}$. If $\mathcal{M}$ is the same for all active learning strategies, we can slightly abuse the notation $\mathcal{A}^{(k)} = \mathcal{M}_{\mathcal{A}^{(k)}}$ to emphasize the focus of active learning algorithms. We also assume that the class probabilities are given by a ground truth GCN; i.e., there exists a GCN $\mathcal{M}^*$ that predicts $\Pr[y_i = c]$ on the entire training set.

Our goal is to minimize the loss under a given budget $b$ at the $k$-th iteration:

$$\min_{\theta^{(k)}, \mathcal{Q}_b} \mathcal{L}^{(k)} \tag{1}$$

# 4 ALIN FRAMEWORK

In this section, we represent ALIN, a comprehensive solution designed to address the challenge of active learning within the context of an incomplete graph. The process underlying our framework can be dissected into two principal components: the query phase and the training phase, both composed of distinct functions as follows:

- Query Phase: This phase encompasses node selection and subsequent updates. During the initial node selection, we utilize the InitNodes function. In subsequent iterations, we calculate node scores, edge scores and combine them to identify the most informative node. We then update the selected nodes and the lost edges associated with them.

- Training Phase: In this phase, we focus on the core of our methodology: a unified loss function that combines node classification and link loss prediction.

*Example 1.* A schematic overview of our proposed ALIN framework is visualized in Fig. 1, where we depict the initial two iterations of the ALIN framework. In first iteration, $(\mathbf{X}, \tilde{\mathbf{Y}}^{(0)}, \tilde{\mathcal{E}}^{(0)})$ contains eight unlabeled nodes and six missing edges. During the Query Phase, two nodes, namely 4 and 5 (highlighted by yellow circles), are chosen for labeling. Consequently, the three edges connected to these nodes are integrated, resulting in the updated

$(\mathbf{X}, \tilde{\mathbf{Y}}_u^{(0)}, \tilde{\mathcal{E}}_u^{(0)})$. Subsequently, $(\mathbf{X}, \tilde{\mathbf{Y}}_u^{(0)}, \tilde{\mathcal{E}}_u^{(0)})$ undergoes the GNN model to generate node embeddings and to predict both $\hat{\mathbf{Y}}^{(0)}$ and $\hat{\mathcal{E}}^{(0)}$. From $(\tilde{\mathbf{Y}}_u^{(0)}, \tilde{\mathcal{E}}_u^{(0)})$ and $(\hat{\mathbf{Y}}^{(0)}, \hat{\mathcal{E}}^{(0)})$, we compute $l_{NC}$ and $l_{LP}$ and its amalgamation $\mathcal{L}^{(0)}$, which is subsequently utilized in the backpropagation process into the GNN. Moving on to the second iteration, we obtain $(\tilde{\mathbf{Y}}^{(1)}, \tilde{\mathcal{E}}^{(1)}) \leftarrow (\tilde{\mathbf{Y}}_u^{(0)}, \tilde{\mathcal{E}}_u^{(0)})$ and two additional nodes, namely 2 and 3 (highlighted by yellow circles), are selected for labeling. As a result, the two associated edges are incorporated, leading to the update of $(\mathbf{X}, \tilde{\mathbf{Y}}_u^{(1)}, \tilde{\mathcal{E}}_u^{(1)})$.

The technical details of the two phases are described in the following.

## 4.1 QUERY PHASE

### 4.1.1 InitNodes

As our framework relies on hidden representations of nodes or the predicted class distribution from the initial model, we operate within iterative settings, necessitating an initial model trained with the seed set. Consequently, we require the 'InitNodes' function to select the initial set of nodes. This function allows us to employ various selection strategies, such as random selection or employing recent methods such as FeatProp [Wu et al., 2019], Centrality [Cai et al., 2017], GraphPart [Ma et al., 2022]. For the implementation of ALIN, we employ GraphPart as the 'InitNodes' function. GraphPart works by dividing the graph into separate partitions and then selecting representative nodes within each partition for active learning with GNN. To mitigate interference across partitions in GraphPart, Ma et al. [2022] also introduces GraphPartFar, a method that penalizes selecting nodes close to medoids chosen in prior partitions, thereby promoting diversity among the returned nodes. According to two variants of GraphPart, we also present another variant of ALIN, termed ALINFar. The distinction between ALIN and ALINFar lies in their respective utilization of the 'InitNodes' function, i.e., ALIN employs the GraphPart function for this purpose, whereas ALINFar opts for the utilization of GraphPartFar as the initialization function.

### 4.1.2 Combine Score

In the context of querying the constituents of the graph, given a prescribed number of $b$ queries, the query function is constructed based on an equilibrium criterion encompassing the informational value of nodes into node score $\phi_{NS}^{(k)}$ and the informational value of edges into edge score $\phi_{ES}^{(k)}$. The amalgamation of $\phi_{NS}^{(k)}$ and $\phi_{ES}^{(k)}$ with a weight parameter denoted as $\alpha$ yields the composite score $\phi_{CS}^{(k)}$.

*Node Score.* The use of entropy as a scoring metric provides valuable insights into the confidence of the GNN model's

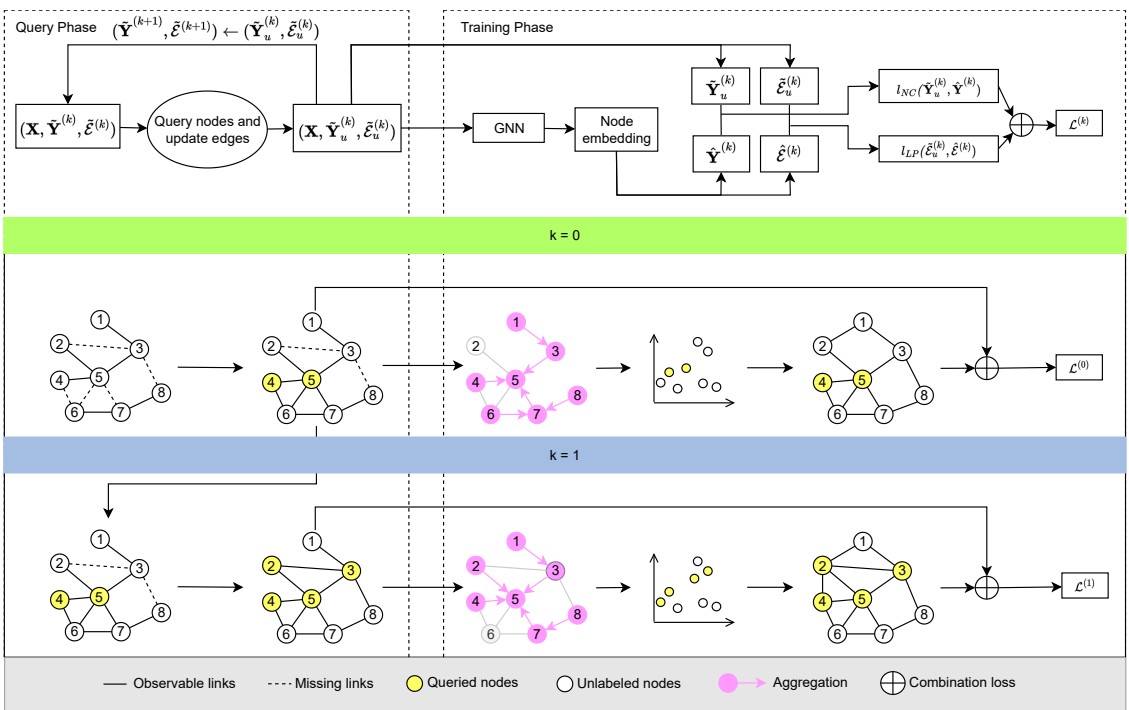

Figure 1: A schematic overview of our proposed ALIN framework.

predictions for individual nodes. Higher entropy values indicate greater uncertainty, suggesting that a node's classification is less certain and may require further exploration or refinement in subsequent iterations of the process. In contrast, lower entropy values signify a higher level of confidence in the node's classification, making it less likely to be selected for additional query iterations. Following by Cai et al. [2017], $\phi_{NS}$ of candidate node $v_i$ at the $k$-th iteration is calculated as follows:

$$\phi_{NS}^{(k)}(v_i) = -\sum_{c=1}^{C} \mathbb{M}_{ic}^{(k)} \log \mathbb{M}_{ic}^{(k)}, \quad (2)$$

where $\mathbb{M}_{ic}^{(k)} = \mathbb{P}(\tilde{\mathbf{Y}}_{ic}^{(k)} = 1|\tilde{\mathcal{G}}^{(k)}, \tilde{\mathbf{Y}}^{(k)}, \mathbf{X})$ is the probability of node $v_i$ belonging to class $c$ predicted by GNN at the $k$-th iteration; $\tilde{\mathbf{Y}}_{ic}^{(k)} = 1$ indicates node $i$ has label $c$. Furthermore, the efficiency of our entropy-based node scoring approach plays a crucial role in accelerating the overall query process. With reduced computational overhead, our framework facilitates faster exploration of the graph and enhances the overall efficiency of the active learning framework.

*Edge Score.* In scenarios where the graph is incomplete, we aim to not only select nodes with high entropy but also nodes to allow the model to learn on a more complete graph. Intuitively, nodes with a larger difference in observable and predicted degrees are prioritized for inclusion in the active learning process since they offer the potential for improving the overall graph representation and classification performance. Thus, $\phi_{ES}$ of node $v_i$ at the $k$-th iteration is

calculated as follows:

$$\phi_{ES}^{(k)}(v_i) = \sum_{n=1}^{N} \mathbb{P}(\tilde{\mathcal{E}}_{in}^{(k)} = 1|\tilde{\mathcal{G}}^{(k)}, \tilde{\mathcal{E}}^{(k)}, \mathbf{X}) - D_{v_i}(\tilde{\mathcal{E}}^{(k)}),$$

$$(3)$$

where $\sum_{n=1}^{N} \mathbb{P}(\tilde{\mathcal{E}}_{in}^{(k)} = 1|\tilde{\mathcal{G}}^{(k)}, \tilde{\mathcal{E}}^{(k)}, \mathbf{X})$ is the probability of node $v_i$ has connect to node $v_n$ and $D_{v_i}(\tilde{\mathcal{E}}^{(k)})$ is the degree of node $v_i$ in $\tilde{\mathcal{E}}^{(k)}$; $\tilde{\mathcal{E}}_{in}^{(k)} = 1$ indicates node $i$ has a connection to node $n$ at the $k$-th. Intuitively, $\phi_{ES}^{(k)}(v_i)$ can be interpreted as the residual degree of a node $v_i$. By incorporating the $\phi_{ES}^{(k)}$ alongside the $\phi_{NS}^{(k)}$, our active learning framework ensures the selection of nodes that not only exhibit uncertainty but also contribute to enhancing the graph's completeness and discriminative power.

The combination of $\phi_{NS}^{(k)}$ and $\phi_{ES}^{(k)}$ represents a promising direction for active learning in graph-based settings. By leveraging uncertainty and graph completeness, our approach strikes a balance between exploration and exploitation, thereby achieving efficient and reliable active learning in real-world scenarios. The versatility of our approach makes it well-suited for a wide range of applications, including social networks, recommendation systems, and bioinformatics, among others

## 4.2 TRAINING PHASE

GNN model training: In our active learning framework, we utilize two loss functions during the training of the Graph

Neural Network (GNN) model. For node classification, we employ the cross entropy loss function, and for link prediction, we utilize the binary cross entropy with logits loss function. The rationale behind using a combined loss is that both $\phi_{NS}^{(k)}$ and $\phi_{ES}^{(k)}$ are influenced by the predictive power of the GNN model. By simultaneously retraining both the node classifier and the edge classifier models, we ensure that the combined score captures valuable information about both node attribute information and the structural information surrounding each node. This concurrent retraining of the edge classifier is motivated by the objective to query nodes that offer a high value of information for both aspects, thereby enhancing the overall representation of the graph. During the model retraining phase, we update the GNN model based on the newly acquired information from the queried nodes. This iterative improvement of the GNN model helps refine the parameters $\theta^{(k)}$ with each query round, resulting in an increasingly accurate and informative model.

## 4.3 THEORETICAL ANALYSIS

Recall that we use $(\mathcal{A}^{(k)})_i = (\mathcal{A}(\tilde{\mathcal{G}}^{(k)}, X))_i \in \mathbb{R}^C$ be the prediction for node $i$ under input $\tilde{\mathcal{G}}^{(k)}, X$, and $(\mathcal{A}^{(k)})_{i,c}$ be the $c$-th element of $(\mathcal{A}^{(k)})_i$ (i.e., the prediction for class $c$). $(\mathcal{M}^*)_i \in \mathbb{R}^C$ is the prediction for node $i$ of ground truth GCN. Our approach shares similarities with the work of Wu et al. [2019] in the context of active learning. However, our framework diverges significantly, particularly in how we handle the incomplete graph structure. We focus on the uncertainty of node, which offers a more nuanced understanding of the incomplete graph compared to simply relying on translated features. The concept of 'translated features' in Wu et al. [2019] refers to pairwise distances between hidden representations of nodes. These features are highly informative when computed on complete graphs, as they encapsulate the full relational structure and interactions between nodes. In a complete graph, every possible link is present, allowing for a comprehensive and accurate representation of node relationships through these features. However, this approach encounters significant challenges in the context of incomplete graphs, where some links are missing. In such scenarios, the absence of certain links can lead to a distorted or incomplete understanding of the node relationships, and translated features no longer accurately represent the actual structure of the network. Our methodology is designed to address this limitation by focusing on a probability-based approach rather than relying solely on translated features. This allows for more accurate and reliable analysis in scenarios where the graph structure is incomplete, ensuring that our conclusions are robust even in the face of missing data.

At $k$-th iteration, Theorem 1 formally shows that choosing the most uncertain nodes can lead to a low node classification loss.

To understand Theorem 1, we note that the first term is the selection of an uncertain node $j \in \mathcal{Q}_b^{(k)}$, and the second term quickly decays with $n$, where $n$ is the total number of nodes in graph $\mathcal{G}$. Therefore, the node classification loss of $\mathcal{A}^{(k)}$ on the graph $\tilde{\mathcal{G}}^{(k)}$ is mostly dependent on the selection of an uncertain node. The assumptions we made in Theorem 1 are pretty standard in the literature, and we illustrate the details in the appendix.

## 5 EXPERIMENTS

### 5.1 EXPERIMENT SETUP

***Dataset.*** To compare against state-of-the-art methods, we experiment on 4 benchmark datasets, including citation networks Citeseer, Cora, and Pubmed [Sen et al., 2008] and co-authorship networks [Shchur et al., 2018]. The summary statistics of the datasets are provided in Table 2. The homophily ratio is defined following Zhu et al. [2020]. In the experimental range with simulated conditions, we remove 30% of the total edges to create incomplete graphs.

***GNN Models.*** We perform experiments over three popular GNN models, including a 3-layer GCN [Kipf and Welling, 2016] with hidden neurons are 128 and 64, respectively, a 3-layer GraphSAGE [Hamilton et al., 2017] with hidden neurons are 128 and 64, respectively, and an 8 attention head-GAT [Hamilton et al., 2017] with 2 hidden layers of size 16 and 8, respectively. To train each model, we use an Adam optimizer with an initial learning rate of $1 \times 10^{-2}$ and weight decay of $5 \times 10^{-4}$. As in the active learning setup, there should not be enough labeled samples to be used as a validation set, we train the GNN model with fixed 200 epochs in all the experiments and evaluate over the full graph.

***Competitive methods.*** We compare active learning methods that can be applied to the iterative setting, divided into two categories: 1) general-purpose methods that are agnostic to the graph structure, namely Random, Density, Uncertainty, and CoreSet; and 2) methods tailored for graph-structured data, including Centrality, AGE, FeatProp, GraphPartFar, ALINFar.

- **Random**: Randomly chooses nodes without any specific criteria.

- **Density** [Cai et al., 2017]: Initially applies clustering to the hidden representations of nodes. It then selects nodes with the highest density score, which is roughly inversely related to the $l_2$-distance between each node and its respective cluster center.

- **Uncertainty** [Settles and Craven, 2008]: Selects nodes with the highest entropy in their predicted class distribution.

- **CoreSet** [Sener and Savarese, 2018]: Utilizes K-Center

**Theorem 1.** *Suppose that the label vector $\tilde{Y}_u^{(k)}$ is sampled independently from the distribution $y_v \sim \eta(v)$ and the loss function $l_{NC}^{(k)}$ is bounded by $[-L, L]$. Then under mild assumptions, there exists a probability $1 - \delta$ the expected classification loss of $\mathcal{A}^{(k)}$ satisfies*

$$\frac{1}{n} l_{NC}^{(k)}(\mathcal{A}^{(k)}|\tilde{\mathcal{G}}^{(k)}, X, \tilde{Y}_u^{(k)}) \leq \sum_{i=1}^{n} \sum_{c=1}^{C} \left[ \frac{\lambda}{n}(\mathcal{M}^*)_{j,c} \min_{j \in \mathcal{Q}_b^{(k)}} |(\mathcal{A}^{(k)})_{i,c} - (\mathcal{A}^{(k)})_{j,c}| + \frac{L}{n}((\mathcal{M}^*)_{i,c} - (\mathcal{M}^*)_{j,c}) \right] + \sqrt{\frac{L \log(1/\delta)}{2n}}$$

(4)

| Dataset | #Nodes | #Edges | #Features | #Classes (C) | Homophily |
|---------|--------|--------|-----------|--------------|-----------|
| Cora | 2,708 | 5,278 | 1,433 | 7 | 0.810 |
| Citeseer | 3,327 | 4,552 | 3,703 | 6 | 0.736 |
| Pubmed | 19,717 | 44,324 | 500 | 3 | 0.802 |
| Coauthor-CS | 18,333 | 81,894 | 6,805 | 6 | 0.808 |

Table 2: Summary statistics of datasets.

clustering on the hidden representations of nodes. Given the scalability issues of the MIP optimized version, a time-efficient greedy approximation, as described in the original work, is used.

- **Centrality**: Chooses nodes with the highest values in graph centrality metrics. Notably, this approach only considers the graph structure and does not take into account node features. Empirical evidence from [Cai et al., 2017] suggests that **Degree** centrality and **PageRank** centrality tend to outperform other metrics and thus we employ **Degree** and **PageRank** as two baselines for comparison.

- **AGE** [Cai et al., 2017]: Quantifies the informativeness of nodes by linearly combining three metrics: centrality, density, and uncertainty. It then selects nodes with the highest combined scores.

- **FeatProp** [Wu et al., 2019]: First conducts K-Means clustering on the aggregated node features and subsequently selects nodes that are closest to the cluster centers.

- **GraphPart** and **GraphPartFar** [Ma et al., 2022]: First obtains a K-partition of a graph using the Clauset-Newman-Moore greedy modularity maximization method [Clauset et al., 2004]. In each part, cluster on the aggregated node features and then choose the nodes closest to the cluster centers.

Following Wu et al. [2019], we evaluate each baseline with a series of label budgets and report the Macro-F1 performance for node classification over the full graph. We note that the results are the average of repeated experiments with 3 random seeds

## 5.2 EXPERIMENT RESULTS ON GCN

The performance comparison between all competitive methods is presented in Table 3. Note-worthy findings are summarized as follows:

- Our proposed ALIN and ALINFar, substantially outperform baseline methods across various budget constraints. Notably, these improvements persist until performance plateaus.

- On smaller datasets like Cora and Citeseer, where the number of nodes and edges is relatively modest, our proposed framework exhibits remarkable superiority, surpassing baseline methods by a notable margin, typically around 1-1.5%. This enhanced performance can be attributed to the framework's adept utilization of feedback from the training process.

- For datasets with more extensive node and edge counts, such as Pubmed and Coauthor-CS, ALIN demonstrates clear advantages over baseline methods, particularly outperforming GraphPart, the state-of-the-art method. Notably, on larger datasets like Pubmed and Coauthor-CS, ALIN proves most effective with smaller budgets, typically around 200-230. However, with a more substantial budget (260), AGE edges slightly ahead of ALIN by approximately 0.2-0.5%, as larger budgets tend to lead to performance saturation.

- GraphPartFar exhibits commendable performance on Cora, trailing ALINFar by a mere 1-2%. However, as datasets expand in size, the loss of numerous edges affects partitioning significantly. Consequently, on Pubmed and Coauthor-CS, GraphPartFar lags behind ALINFar by approximately 4-5%.

- In the context of the second-best performing methods, both ALINFar and Uncertainty shine on Cora and Citeseer. However, on Pubmed and Coauthor-CS, Uncer-

| Baselines | Cora | | | Citeseer | | |
|---|---|---|---|---|---|---|
| Buget | 200 | 230 | 260 | 200 | 230 | 260 |
| Random | $76.6 \pm 0.8$ | $78.6 \pm 2.2$ | $79.4 \pm 0.5$ | $61.8 \pm 0.3$ | $61.0 \pm 0.4$ | $63.1 \pm 1.4$ |
| Density | $73.1 \pm 1.1$ | $74.5 \pm 1.6$ | $76.4 \pm 1.9$ | $61.6 \pm 0.6$ | $60.4 \pm 1.3$ | $58.4 \pm 1.4$ |
| Uncertainty | $78.7 \pm 0.7$ | $\underline{80.6} \pm 0.9$ | $80.5 \pm 1.2$ | $63.3 \pm 1.1$ | $\underline{64.2} \pm 1.3$ | $64.5 \pm 0.7$ |
| CoreSet | $77.9 \pm 1.8$ | $79.1 \pm 0.8$ | $79.9 \pm 0.2$ | $61.2 \pm 0.2$ | $61.7 \pm 1.4$ | $65.8 \pm 0.6$ |
| Degree | $72.2 \pm 0.4$ | $73.5 \pm 0.8$ | $75.7 \pm 1.2$ | $53.9 \pm 1.8$ | $54.9 \pm 1.1$ | $56.4 \pm 1.7$ |
| Pagerank | $77.8 \pm 0.5$ | $78.5 \pm 0.1$ | $79.5 \pm 0.6$ | $63.1 \pm 0.2$ | $63.6 \pm 0.1$ | $64.1 \pm 0.2$ |
| AGE | $77.6 \pm 0.9$ | $77.7 \pm 1.2$ | $79.6 \pm 0.2$ | $63.2 \pm 0.2$ | $64.1 \pm 0.9$ | $66.0 \pm 1.5$ |
| FeatProp | $72.1 \pm 1.9$ | $73.1 \pm 0.8$ | $74.5 \pm 0.6$ | $50.6 \pm 1.3$ | $55.8 \pm 0.5$ | $57.2 \pm 2.2$ |
| GraphPart | $72.8 \pm 1.8$ | $73.7 \pm 1.2$ | $75.1 \pm 0.6$ | $54.1 \pm 0.4$ | $54.8 \pm 1.1$ | $57.2 \pm 1.7$ |
| GraphPartFar | $77.8 \pm 0.3$ | $78.4 \pm 0.5$ | $78.4 \pm 0.6$ | $60.9 \pm 2.0$ | $61.4 \pm 2.1$ | $62.3 \pm 1.7$ |
| ALIN | $\mathbf{79.8} \pm 1.3$ | $80.4 \pm 0.6$ | $\mathbf{81.6} \pm 1.0$ | $\underline{63.4} \pm 1.1$ | $63.9 \pm 1.2$ | $\underline{66.1} \pm 0.9$ |
| ALINFar | $\underline{78.9} \pm 1.2$ | $\mathbf{81.2} \pm 1.1$ | $\underline{81.4} \pm 1.7$ | $\mathbf{64.4} \pm 1.3$ | $\mathbf{64.7} \pm 1.1$ | $\mathbf{66.4} \pm 0.6$ |
| Baselines | Pubmed | | | Coauthor-CS | | |
| Buget | 200 | 230 | 260 | 200 | 230 | 260 |
| Random | $77.1 \pm 0.5$ | $77.5 \pm 1.3$ | $78.8 \pm 1.0$ | $77.8 \pm 0.4$ | $82.6 \pm 2.6$ | $81.3 \pm 3.1$ |
| Density | $77.2 \pm 0.4$ | $76.9 \pm 1.2$ | $77.6 \pm 0.9$ | $79.3 \pm 1.5$ | $75.0 \pm 0.8$ | $79.3 \pm 4.0$ |
| Uncertainty | $77.3 \pm 0.4$ | $78.5 \pm 1.5$ | $79.0 \pm 0.3$ | $84.3 \pm 1.0$ | $85.4 \pm 1.5$ | $85.6 \pm 0.4$ |
| CoreSet | $76.4 \pm 0.9$ | $77.3 \pm 0.8$ | $77.1 \pm 0.7$ | $57.2 \pm 3.1$ | $62.7 \pm 5.2$ | $64.1 \pm 3.8$ |
| Degree | $76.1 \pm 1.2$ | $75.7 \pm 0.5$ | $75.7 \pm 0.5$ | $60.0 \pm 0.2$ | $59.9 \pm 0.1$ | $60.2 \pm 0.5$ |
| Pagerank | $75.8 \pm 0.8$ | $76.2 \pm 0.2$ | $77.5 \pm 0.4$ | $84.4 \pm 0.7$ | $84.0 \pm 1.6$ | $84.3 \pm 1.0$ |
| AGE | $\underline{78.3} \pm 1.0$ | $\underline{79.0} \pm 0.2$ | $\underline{79.6} \pm 0.6$ | $\underline{85.1} \pm 0.3$ | $\underline{85.5} \pm 1.1$ | $\mathbf{86.6} \pm 0.3$ |
| FeatProp | $74.0 \pm 0.8$ | $73.4 \pm 0.7$ | $73.6 \pm 1.4$ | $72.7 \pm 4.7$ | $77.7 \pm 1.8$ | $78.3 \pm 1.0$ |
| GraphPart | $73.8 \pm 0.6$ | $74.5 \pm 0.8$ | $74.2 \pm 0.9$ | $77.0 \pm 3.6$ | $79.0 \pm 2.6$ | $80.9 \pm 2.4$ |
| GraphPartFar | $75.1 \pm 0.5$ | $75.1 \pm 0.6$ | $74.8 \pm 0.7$ | $80.2 \pm 1.7$ | $84.9 \pm 1.3$ | $85.0 \pm 0.8$ |
| ALIN | $\mathbf{79.0} \pm 0.8$ | $\mathbf{79.3} \pm 0.5$ | $\mathbf{80.1} \pm 0.6$ | $\mathbf{85.4} \pm 1.2$ | $\mathbf{86.1} \pm 0.7$ | $\underline{85.8} \pm 0.5$ |
| ALINFar | $75.0 \pm 1.6$ | $77.7 \pm 1.7$ | $78.0 \pm 0.9$ | $84.3 \pm 1.1$ | $84.8 \pm 0.3$ | $85.6 \pm 0.5$ |

Table 3: Summary of the performance of GCN on each benchmark. The **bold** marker denotes the best performance and the underlined marker denotes the second-best performance.

tainty's performance is lackluster, with AGE emerging as the second-best performer. The discrepancy arises because Uncertainty operates independently of the graph structure, which becomes problematic for larger datasets where edge loss entails more significant information loss. In contrast, AGE amalgamates centrality, density, and uncertainty scores, proving advantageous in scenarios where data lacks edges in extensive graphs. Consequently, AGE consistently outperforms Uncertainty in these settings.

Overall, our experimental results showcase the robustness and versatility of the ALIN framework and its extensions, shedding light on their adaptability across diverse graph datasets and budget constraints.

## 6 CONCLUSION AND DISCUSSION

In this work, we embarked on a comprehensive exploration of the active learning paradigm tailored specifically for Graph Neural Networks (GNNs) operating on incomplete graphs. Drawing inspiration from the synergy between node and edge information, we introduced a novel framework designed to harness the unique potential inherent in incomplete graph structures. Our experiments yielded compelling evidence of the efficacy of our proposed framework. We demonstrated that it not only outperforms existing state-of-the-art baseline active learning methods but does so consistently across a variety of real-world datasets and scenarios. These findings underscore the pivotal role of our approach in advancing active learning strategies in the context of incomplete graph data.

Several avenues for future research may include: 1) Investigating dynamic edge scoring mechanisms that adapt to the evolving graph structure could be fruitful and 2) Exploring the synergy between our active learning framework and graph generative models could open doors to novel applications.

**Author Contributions**

All authors contributed to the study's conception and design. Cong Tran had the idea for the article. Material prepa-

ration, data collection, and analysis were performed by Tung Khong. Experiments were conducted by Tung Khong. The first draft of the manuscript was written by Tung Khong and all authors commented on previous versions of the manuscript. All authors read and approved the final manuscript.

## Acknowledgements

This work was supported by the Postdoctoral Scholarship Programme of Vingroup Innovation Foundation (VINIF), code VINIF.2023.STS.58, and the research project coded DT. 18/24, funded by the Ministry of Information and Communication, 2024.

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

# ALIN: An Active Learning Framework for Incomplete Networks (Supplementary Material)

**Tung Khong**[1]  **Cuong Pham**[1]  **Cong Tran**[1]

[1]Posts & Telecommunications Institute of Technology, Hanoi, Vietnam

## A  ALIN ALGORITHM

In this section, we describe the ALIN algorithm in detail. ALIN algorithm includes two principal components: the query phase and the training phase.

- Query Phase: This phase encompasses node selection and subsequent updates, detailed in lines 3–15 of Algorithm 1. During the initial node selection, we utilize the InitNodes function (line 4). In subsequent iterations, we calculate node scores (line 8), and edge scores (line 9), and combine them (line 10) to identify the most informative node (line 11). We then update the selected nodes and the lost edges associated with them (lines 13 –15).

- Training Phase: In this phase, we focus on the core of our methodology: a unified loss function that combines node classification and link loss prediction, as described in lines 16–19.

## B  GENERALIZABILITY TO OTHER GNNS

We further present experiment results of all competitive methods on other GNN architectures in Table 4, in which the GCN backbone is replaced by GAT and GraphSAGE accordingly. Our proposed framework consistently demonstrates its capacity to enhance the accuracy of node classification tasks, even when transitioning GAT and GraphSAGE. This substantiates the framework's robust applicability across diverse GNN models.

Notably, among the benchmark methods including Random, Density, CoreSet, Pagerank, GraphPartFar, and ALINFar, CoreSet exhibits exceptional performance on GAT, surpassing GraphSAGE by approximately 4%. In contrast, the other baseline methods exhibit a narrower performance gap of around 1-2% when comparing GAT to GraphSAGE. This superiority of CoreSet on GAT underscores GAT's effectiveness in capturing pertinent information from neighboring nodes, even in scenarios with missing connections, courtesy of its adaptive attention mechanism that judiciously weighs the significance of various neighbors.

Conversely, GraphSAGE, Uncertainty, Degree, and FeatProp consistently outperform GAT, achieving improvements of roughly 0.5-1%. Notably, GraphSAGE is favored for its computational efficiency and scalability, making it well-suited for large graph datasets. Its resilience to the absence of edges between nodes is attributed to its neighbor sampling strategy.

It is noteworthy that both CoreSet and FeatProp rely on K-means clustering based on hidden node representations, which introduces sensitivity to different runs, particularly when applied to a limited number of labels. Consequently, these two baseline methods exhibit a considerably higher standard deviation compared to the other baselines.

## C  HYPERPARAMETER SENSITIVITY

We further the experiment results on tuning hyperparameters, and carry out experiments as follows in Fig. 2, we investigate the impact of hyperparameters $\alpha$, which adjusts the balance between two terms for the query node selection at line 10 of

---

**Algorithm 1** ALIN Algorithm

---

**Input:** $\mathbf{X}$, $\tilde{\mathcal{E}}^{(0)}$, $\tilde{\mathbf{Y}}^{(0)} = \emptyset$, Hyperparameters $(B, \alpha, \beta)$, Trainable model parameters $\theta^{(0)}$, Training iterations $T$, iteration
$\quad$ $k = 0$
**Output:** $\tilde{\mathcal{G}}^{(K)}$, $\theta^{(K)}$

1: **while** $k < K$ **do**
2: $\quad$ $b \leftarrow [B/K]$
3: $\quad$ **if** $k = 0$ **then** $\hfill \triangleright$ Start Query Phase
4: $\quad\quad$ $\mathcal{Q}_b^{(k)} \leftarrow \text{InitNodes}(\tilde{\mathcal{E}}^{(0)}, \mathbf{X})$
5: $\quad$ **else**
6: $\quad\quad$ $\tilde{\mathbf{Y}}^{(k)} \leftarrow \tilde{\mathbf{Y}}_u^{(k-1)}$
7: $\quad\quad$ $\tilde{\mathcal{E}}^{(k)} \leftarrow \tilde{\mathcal{E}}_u^{(k-1)}$
8: $\quad\quad$ $\phi_{NS}^{(k)} \leftarrow$ calculate Node Score for each node following Eq. (2)
9: $\quad\quad$ $\phi_{ES}^{(k)} \leftarrow$ calculate Edge Score for each node following Eq. (3)
10: $\quad\quad$ $\phi_{CS}^{(k)} \leftarrow \alpha \cdot \phi_{NS}^{(k)} + (1 - \alpha) \cdot \phi_{ES}^{(k)}$
11: $\quad\quad$ $\mathcal{Q}_b^{(k)} \leftarrow$ Top $b$ nodes from $\phi_{CS}^{(k)}$
12: $\quad$ **end if**
13: $\quad$ From $\mathcal{Q}_b^{(k)}$ update selected nodes and associated edges from the oracle to $\tilde{\mathbf{Y}}_q^{(k)}$, $\tilde{\mathcal{E}}_q^{(k)}$
14: $\quad$ $\tilde{\mathbf{Y}}_u^{(k)} \leftarrow \tilde{\mathbf{Y}}^{(k)} \cup \tilde{\mathbf{Y}}_q^{(k)}$
15: $\quad$ $\tilde{\mathcal{E}}_u^{(k)} \leftarrow \tilde{\mathcal{E}}^{(k)} \cup \tilde{\mathcal{E}}_q^{(k)}$ $\hfill \triangleright$ End Query Phase
16: $\quad$ **for** $t = 1$ to $T$ **do** $\hfill \triangleright$ Start Training Phase
17: $\quad\quad$ From $\theta^{(k)}$, calculate $\hat{\mathbf{Y}}^{(k)}$ and $\hat{\mathcal{E}}^{(k)}$
18: $\quad\quad$ $\mathcal{L}^{(k)} \leftarrow \beta \cdot l_{NC}(\tilde{\mathbf{Y}}_u^{(k)}, \hat{\mathbf{Y}}^{(k)}) + (1 - \beta) \cdot l_{LP}(\tilde{\mathcal{E}}_u^{(k)}, \hat{\mathcal{E}}^{(k)})$
19: $\quad\quad$ Backpropagation to $\theta^{(k)}$
20: $\quad$ **end for** $\hfill \triangleright$ End Training Phase
21: $\quad$ $k \leftarrow k + 1$
22: **end while**

---

| Baselines | GAT | | | GraphSAGE | | |
|---|---|---|---|---|---|---|
| Buget | 200 | 230 | 260 | 200 | 230 | 260 |
| Random | $78.4 \pm 0.6$ | $78.0 \pm 1.1$ | $79.5 \pm 0.3$ | $76.2 \pm 0.5$ | $78.2 \pm 0.7$ | $78.7 \pm 0.4$ |
| Density | $77.2 \pm 0.9$ | $80.1 \pm 0.3$ | $79.9 \pm 0.4$ | $77.4 \pm 0.4$ | $80.6 \pm 1.5$ | $81.7 \pm 1.6$ |
| Uncertainty | $\underline{79.8 \pm 1.0}$ | $81.3 \pm 1.4$ | $81.5 \pm 1.0$ | $\underline{79.4 \pm 0.7}$ | $\underline{80.8 \pm 1.3}$ | $81.3 \pm 0.7$ |
| CoreSet | $66.5 \pm 1.4$ | $65.7 \pm 2.7$ | $66.1 \pm 3.7$ | $61.0 \pm 4.2$ | $62.5 \pm 2.4$ | $62.1 \pm 3.2$ |
| Degree | $74.7 \pm 1.1$ | $75.3 \pm 1.1$ | $76.8 \pm 0.4$ | $76.4 \pm 0.7$ | $76.7 \pm 0.6$ | $79.1 \pm 0.6$ |
| Pagerank | $77.7 \pm 0.5$ | $78.9 \pm 0.3$ | $80.3 \pm 0.6$ | $77.6 \pm 0.3$ | $77.9 \pm 0.5$ | $80.1 \pm 0.7$ |
| AGE | $78.8 \pm 0.1$ | $79.9 \pm 0.5$ | $80.7 \pm 0.1$ | $77.3 \pm 1.6$ | $80.0 \pm 0.4$ | $80.8 \pm 0.4$ |
| FeatProp | $72.2 \pm 0.7$ | $73.8 \pm 0.4$ | $75.9 \pm 0.4$ | $73.1 \pm 0.6$ | $75.0 \pm 0.8$ | $76.1 \pm 1.2$ |
| GraphPart | $72.8 \pm 0.9$ | $74.3 \pm 0.9$ | $75.1 \pm 0.3$ | $74.4 \pm 0.7$ | $74.7 \pm 1.0$ | $75.0 \pm 0.7$ |
| GraphPartFar | $77.9 \pm 0.4$ | $76.9 \pm 0.6$ | $78.6 \pm 0.8$ | $76.7 \pm 0.9$ | $76.8 \pm 0.5$ | $77.6 \pm 0.1$ |
| ALIN | $\mathbf{80.5} \pm 0.9$ | $\underline{81.4 \pm 1.2}$ | $\underline{81.8 \pm 0.7}$ | $79.0 \pm 0.5$ | $79.9 \pm 1.8$ | $\underline{81.6 \pm 1.4}$ |
| ALINFar | $79.7 \pm 0.9$ | $\mathbf{82.1} \pm 0.2$ | $\mathbf{82.7} \pm 0.8$ | $\mathbf{79.9} \pm 0.8$ | $\mathbf{81.7} \pm 0.3$ | $\mathbf{82.4} \pm 0.4$ |

Table 4: Summary of the performance of others GNN on Cora dataset. The **bold** marker denotes the best performance and the underlined marker denotes the second-best performance.

the Algorithm 1, on the performance of both ALIN and ALINFar when $B = 230$ and $K = 8$ on Cora dataset. In these

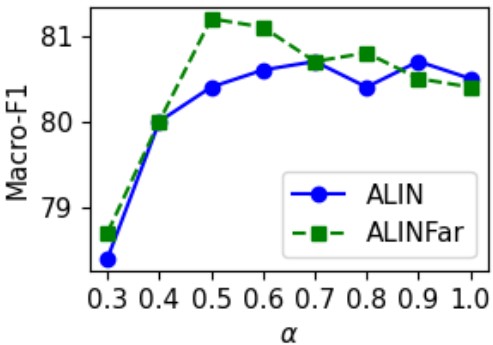

Figure 2: The expected influence by ALIN and ALINFar according to different values hyperparameters $\alpha$

experiments, we only show the results from the GCN model since those from the other GNN models follow a similar trend. We observe the following:

- The case of $\alpha = 0.5$ exhibits the best performance almost consistently.
- When $\alpha = 1$, ALIN and ALINFar only depend on the node score, resulting in low performance.
- Setting $\alpha$ to 0.3 leads to much lower performance as this setting overemphasizes the edge score, detracting from our ultimate goal of node classification.

These findings emphasize the delicate interplay between node uncertainty, edge information, and the overarching goal of accurate node classification within the ALIN framework.

## D   WEIGHT GROWTH OF FUNCTION

In this section, we delve into the intricate dynamics of $\beta$ and its evolution across epochs, achieved through the utilization of a growth function. Recall that $\beta$ is the weight of combined loss at line 18 of Algorithm 1, which plays a pivotal role in balancing the trade-off between two essential tasks: optimal link prediction and the ultimate goal of node classification.

At the outset of training, during the initial epochs, we set $\beta$ to a value of 0.05. This choice steers the model's focus primarily towards solving the optimal link prediction problem. In contrast, as we approach the final epochs, our objective is to set $\beta$ to 1, emphasizing the model's commitment to the ultimate task of node classification.

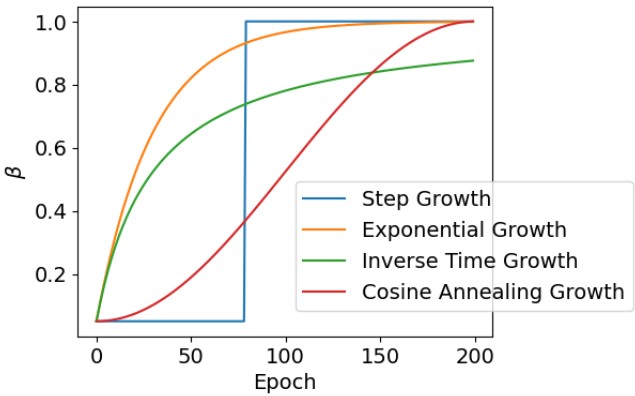

Figure 3: Weight growth functions

Achieving this gradual transition in $\beta$ necessitates the implementation of a suitable growth function. We explored various weight growth functions to identify the most effective approach, weight growth functions shown in Fig. 3. The results of these experiments are summarized in Table 5, revealing distinct performance characteristics among different growth functions.

Notably, the Cosine Annealing, Step, and Exponential growth functions emerge as superior choices when compared to the Inverse Time growth function. Both the Cosine Annealing and Step growth functions exhibit an advantageous pattern of gradually increasing the $\beta$ parameter during the middle epochs. This characteristic aligns seamlessly with the requirements of the Link Prediction task, which thrives on sustained training over multiple epochs, rather than experiencing a premature reduction in emphasis. Furthermore, the Exponential growth function proves notable for its ability to swiftly approach a $\beta$ value close to 1 during the latter epochs. This rapid convergence to a higher $\beta$ value positions the Exponential growth function as a compelling choice, outperforming the Cosine Annealing growth function in terms of accuracy.

In summary, our approach to the weight growth of $\beta$ involves a thoughtful selection of growth functions, ultimately tailored to strike the right balance between optimizing link prediction and achieving robust node classification. The choice of growth function is a critical aspect of our framework, as it ensures that the model evolves and adapts its focus in a manner that aligns with the evolving requirements of the tasks at hand.

| Weight Growth Function | Baselines | Budget | | |
|---|---|---|---|---|
| | | 200 | 230 | 260 |
| Step Growth | ALIN | **79.8** $\pm$ 1.3 | $\underline{80.4}$ $\pm$ 0.6 | **81.6** $\pm$ 1.0 |
| | ALINFar | $\underline{78.9}$ $\pm$ 1.2 | **81.2** $\pm$ 1.1 | $\underline{81.4}$ $\pm$ 1.7 |
| Inverse Time Growth | ALIN | 63.3 $\pm$ 1.7 | 62.3 $\pm$ 2.3 | 65.2 $\pm$ 0.6 |
| | ALINFar | 63.0 $\pm$ 2.1 | 64.3 $\pm$ 1.7 | 65.6 $\pm$ 0.4 |
| Exponential Growth | ALIN | 78.2 $\pm$ 0.5 | 77.6 $\pm$ 1.5 | 79.7 $\pm$ 0.6 |
| | ALINFar | 78.6 $\pm$ 1.5 | 77.7 $\pm$ 1.1 | 80.1 $\pm$ 0.5 |
| Cosine Annealing Growth | ALIN | 76.0 $\pm$ 0.6 | 78.1 $\pm$ 0.8 | 76.8 $\pm$ 2.0 |
| | ALINFar | 74.7 $\pm$ 1.8 | 75.8 $\pm$ 0.4 | 76.7 $\pm$ 0.7 |

Table 5: Summary of the performance of weight growth functions using GNN on Cora dataset. The numerical values represent the average Macro-F1 score of 3 independent trials. The **bold** marker denotes the best performance and the underlined marker denotes the second-best performance.

# E  PROOF OF THEOREM 1

Our approach shares similarities with the work of Wu et al. [2019]. For simplicity, for any model $\mathcal{M}^{(k)}$ at $k$-th iteration let $(\mathcal{M}^{(k)})_i = (\mathcal{M}(\tilde{\mathcal{G}}^{(k)}, X))_i \in \mathbb{R}^C$ be the prediction for node $i$ under input $\tilde{\mathcal{G}}^{(k)}, X$, and $\mathcal{M}_{i,c}^{(k)}$ be the $c$-th element of $(\mathcal{M}^{(k)})_i$ (i.e., the prediction for class $c$). We also make the following assumptions:

**Assumption 1.** We assume that $\mathcal{A}^{(k)}$ overfits to the training data. Specifically, we also assume the following two conditions: i) $\mathcal{A}^{(k)}$ attains zero training loss on the set $\mathcal{Q}_b^{(k)}$, and ii) for any unlabeled data pair $(x_i, x_j)$ where $i \notin \mathcal{Q}_b^{(k)}$ and $j \in \mathcal{Q}_b^{(k)}$, it holds that $(\mathcal{A}^{(k)})_{i,y_j} \leq (\mathcal{A}^{(k)})_{j,y_j}$ and $(\mathcal{A}^{(k)})_{i,c} \geq (\mathcal{A}^{(k)})_{j,c}$ for all $c \neq y_j$. The second condition implies that $\mathcal{A}^{(k)}$ achieves low confidence on unseen samples and high confidence on trained samples. Additionally, we assume that the class probabilities are determined by a ground truth GCN, denoted as $\mathcal{M}^*$, which predicts $\Pr[y_i = c]$ for the entire training set. In the literature, this is a common assumption. Both $\mathcal{A}^{(k)}$ and $\mathcal{M}^*$ calculate probability outputs.

**Assumption 2.** We assume that $l_{NC}$ bounded in $[-L, L]$ is Lipschitz with constant $\lambda$. The loss function is naturally Lipschitz for many common loss functions such as mean squared error, hinge loss, and cross-entropy when the model output is constrained within certain bounds. This assumption finds frequent application in deep learning theory (e.g., [Allen-Zhu et al., 2017, Du et al., 2019]).

**Assumption 3.** We assume that ReLU function activates with probability $1/2$. This assumption, frequently made in the analysis of neural network loss surfaces, is also used in [Choromanska et al., 2015, Kawaguchi, 2016, Xu et al., 2018]. It is consistent with practical observations where, typically, approximately half of the ReLU neurons can activate.

With these assumptions in place, we can prove Theorem 1

**Theorem 1** (restated). *Suppose Assumptions 1-3 hold, and the label vector $\tilde{Y}_u^{(k)}$ is sampled independently from the distribution $y_v \sim \eta(v)$ for every $v \in V$. Then with probability $1 - \delta$ the expected classification loss of $\mathcal{A}^{(k)}$ satisfies*

$$\frac{1}{n} l_{NC}^{(k)}(\mathcal{A}^{(k)} | \tilde{\mathcal{G}}^{(k)}, X, \tilde{Y}_u^{(k)}) \leq \sum_{i=1}^{n} \sum_{c=1}^{C} \left[ \frac{\lambda}{n} (\mathcal{M}^*)_{j,c} \min_{j \in \mathcal{Q}_b^{(k)}} |(\mathcal{A}^{(k)})_{i,c} - (\mathcal{A}^{(k)})_{j,c}| + \frac{L}{n}((\mathcal{M}^*)_{i,c} - (\mathcal{M}^*)_{j,c}) \right] + \sqrt{\frac{L \log(1/\delta)}{2n}} \tag{5}$$

*Proof.* Consider the following random process: Fix $y_j$ for $j \in \mathcal{Q}_b^{(k)}$ and therefore the resulting model $\mathcal{A}^{(k)}$, and suppose the (hidden) labels $y_i$ for $i \notin \mathcal{Q}_b^{(k)}$ is randomly sampled according to $\eta(v_i)$. Let $i \in V \setminus \mathcal{Q}_b^{(k)}$ be any node and $j \in \mathcal{Q}_b^{(k)}$. We have

$$\begin{aligned}
\mathbb{E}_{y \sim \eta(i)} \left[ l_{NC}((\mathcal{A}^{(k)})_i, y) \right] &= \sum_{c=1}^{C} \Pr[y_i = c] l_{NC}((\mathcal{A}^{(k)})_{i,c}, c) \\
&= \sum_{c=1}^{C} \Pr[y_j = c] l_{NC}((\mathcal{A}^{(k)})_{i,c}, c) + \sum_{c=1}^{C} (\Pr[y_i = c] - \Pr[y_j = c]) l_{NC}((\mathcal{A}^{(k)})_{i,c}, c).
\end{aligned} \tag{6}$$

For the first term, we have

$$\begin{aligned}
\sum_{c=1}^{C} \Pr[y_j = c] l_{NC}((\mathcal{A}^{(k)})_{i,c}, c) &= \sum_{c=1}^{C} \Pr[y_j = c] \left[ l_{NC}((\mathcal{A}^{(k)})_{i,c}, c) - l_{NC}((\mathcal{A}^{(k)})_{j,c}, c) \right] \\
&\quad + \sum_{c=1}^{C} \Pr[y_j = c] l_{NC}((\mathcal{A}^{(k)})_{j,c}, c) \\
&= \sum_{c=1}^{C} \Pr[y_j = c] \left[ l_{NC}((\mathcal{A}^{(k)})_{i,c}, c) - l_{NC}((\mathcal{A}^{(k)})_{j,c}, c) \right] \\
&\leq \lambda \sum_{c=1}^{C} \Pr[y_j = c] |(\mathcal{A}^{(k)})_{i,c} - (\mathcal{A}^{(k)})_{j,c}|
\end{aligned} \tag{7}$$

The last inequality holds from the Lipschitz continuity of $l$. Now from Assumption 1, we have $(\mathcal{A}^{(k)})_{i,c} \geq (\mathcal{A}^{(k)})_{j,c}$ for $c \neq y_j$ and $(\mathcal{A}^{(k)})_{i,c} \leq (\mathcal{A}^{(k)})_{j,c}$ otherwise.

Now for the second loss in Eq. (6) we use the property that $\mathcal{M}^*$ computes the ground truth:

$$
\sum_{c=1}^{C}(\Pr[y_i = c] - \Pr[y_j = c])l_{NC}((\mathcal{A}^{(k)})_{i,c}, c) = \sum_{c=1}^{C}((\mathcal{M}^*)_{i,c} - (\mathcal{M}^*)_{j,c})l_{NC}((\mathcal{A}^{(k)})_{i,c}, c)
$$
$$
\leq \sum_{c=1}^{C} L((\mathcal{M}^*)_{i,c} - (\mathcal{M}^*)_{j,c}). \tag{8}
$$

The last inequality follows from $l \in [-L, L]$.

Combining the two parts to Eq. (6), we obtain

$$
\mathbb{E}_{y \sim \eta(i)} \left[ l_{NC}((\mathcal{A}^{(k)})_i, y) \right] \leq \lambda \sum_{c=1}^{C} \Pr[y_j = c] |(\mathcal{A}^{(k)})_{i,c} - (\mathcal{A}^{(k)})_{j,c}| + \sum_{c=1}^{C} L((\mathcal{M}^*)_{i,c} - (\mathcal{M}^*)_{j,c})
$$
$$
\leq \sum_{c=1}^{C} \left[ \lambda(\mathcal{M}^*)_{j,c} |(\mathcal{A}^{(k)})_{i,c} - (\mathcal{A}^{(k)})_{j,c}| + L((\mathcal{M}^*)_{i,c} - (\mathcal{M}^*)_{j,c}) \right] \tag{9}
$$

To minimize the right-hand side (RHS) of Eq. (9), it is necessary that both $|(\mathcal{A}^{(k)})_{i,c} - (\mathcal{A}^{(k)})_{j,c}|$ and $((\mathcal{M}^*)_{i,c} - (\mathcal{M}^*)_{j,c})$ are minimized.

- To achieve the minimum of $|(\mathcal{A}^{(k)})_{i,c} - (\mathcal{A}^{(k)})_{j,c}|$, consider assumption $(\mathcal{A}^{(k)})_i$ represents the $C$-dimensional output vector of model $(\mathcal{A}^{(k)})$ for node $i$. Given that $(\mathcal{A}^{(k)})_i$ indicates uncertainty in $V \setminus \mathcal{Q}_b^{(k)}$, the elements of the output vector are relatively similar. Therefore, selecting node $j$ as the central node of all $(\mathcal{A}^{(k)})$ outputs where the output vector elements are most alike ensures that node $j$ embodies the highest uncertainty.

- For minimizing $((\mathcal{M}^*)_{i,c} - (\mathcal{M}^*)_{j,c})$, note that $\mathcal{M}^*$ exhibits certainty with unseen data. As an ideal model, $\mathcal{M}^*$ accurately represents the underlying class probabilities and does not overfit specific training samples. Its calibrated confidence ensures that its predictions' confidence levels are consistent with the actual likelihood, for both seen and unseen data. Consequently, selecting any node $j$ does not impact the value of $((\mathcal{M}^*)_{i,c} - (\mathcal{M}^*)_{j,c})$.

Therefore, by selecting node $j$ with the highest uncertainty in Eq. (9), we obtain:

$$
\mathbb{E}_{y \sim \eta(i)} \left[ l_{NC}((\mathcal{A}^{(k)})_i, y) \right] \leq \sum_{c=1}^{C} \left[ \lambda(\mathcal{M}^*)_{j,c} \min_{j \in \mathcal{Q}_b^{(k)}} |(\mathcal{A}^{(k)})_{i,c} - (\mathcal{A}^{(k)})_{j,c}| + L((\mathcal{M}^*)_{i,c} - (\mathcal{M}^*)_{j,c}) \right] \tag{10}
$$

Now notice that

$$
l_{NC}(\mathcal{A}^{(k)}|G, X, Y) = \sum_{i \in V \setminus \mathcal{Q}_b^{(k)}} l_{NC}((\mathcal{A}^{(k)})_i, y_i) + \sum_{j \in \mathcal{Q}_b^{(k)}} l_{NC}((\mathcal{A}^{(k)})_j, y_j) = \sum_{i \in V \setminus \mathcal{Q}_b^{(k)}} l_{NC}((\mathcal{A}^{(k)})_i, y_i). \tag{11}
$$

Consider the following process: we first get $\tilde{\mathcal{G}}^{(k)}, X$ as input, which induces $\eta(i)$ for $i \in [n]$. Note that $\mathcal{M}^*$ gives the ground truth $\eta(i)$ for every $i$ so distributions $\eta(i) \equiv \eta_{X,G}(i)$. Then the algorithm $\mathcal{A}^{(k)}$ chooses the set $\mathcal{Q}_b^{(k)}$ to label. After that, we randomly sample $y_j \sim \eta(j)$ for $j \in \mathcal{Q}_b^{(k)}$ and use the labels to train model $\mathcal{A}^{(k)}$. At last, we randomly sample $y_i \sim \eta(i)$ and obtain loss $l_{NC}^{(k)}(\mathcal{A}^{(k)}|\tilde{\mathcal{G}}^{(k)}, X, \tilde{\mathbf{Y}}_u^{(k)})$. Note that the sampling of all $y_i$ for $i \in V \setminus \mathcal{Q}_b^{(k)}$ is after we fix the model $\mathcal{A}^{(k)}$, and knowing exact values of $y_j$ for $j \in \mathcal{Q}_b^{(k)}$ does not give any information of $y_i$ (since $\eta(i)$ is only determined by $\tilde{\mathcal{G}}^{(k)}, X$). Now we use Hoeffding's inequality (Theorem 2) with $Z_i = l_{NC}(\mathcal{A}^{(k)}|\tilde{\mathcal{G}}^{(k)}, X, \tilde{\mathbf{Y}}_u^{(k)})$; we have $-L \leq Z_i \leq L$ by our assumption, and recall that $|V \setminus \mathcal{Q}_b^{(k)}| = n - b$. Let $\delta$ be the RHS of Eq. (15), we have that with probability $1 - \delta$,

$$
\frac{1}{n-b} \sum_{i \in V \setminus \mathcal{Q}_b^{(k)}} l_{NC}((\mathcal{A}^{(k)})_i, y_i) - \frac{1}{n-b} \mathbb{E}_{y \sim \eta(i), \sigma} \left[ l_{NC}((\mathcal{A}^{(k)})_i, y) \right] \leq \sqrt{\frac{L \log(1/\delta)}{2(n-b)}} \tag{12}
$$

Now plug in Eq. (10), multiply both sides by $(n-b)$ and rearrange. We obtain that

$$
\sum_{i \in V \setminus \mathcal{Q}_b^{(k)}} l_{NC}((\mathcal{A}^{(k)})_i, y_i) \leq \sum_{i \in V \setminus \mathcal{Q}_b^{(k)}} \sum_{c=1}^{C} \left[ \lambda(\mathcal{M}^*)_{j,c} \min_{j \in \mathcal{Q}_b^{(k)}} |(\mathcal{A}^{(k)})_{i,c} - (\mathcal{A}^{(k)})_{j,c}| + L((\mathcal{M}^*)_{i,c} - (\mathcal{M}^*)_{j,c}) \right]
$$
$$
+ \sqrt{\frac{L \log(1/\delta)(n-b)}{2}}
$$
(13)

Now note that since the random draws of $y_i$ are completely irrelevant with the training of $\mathcal{A}^{(k)}$, we can also sample $y_i$ together with $y_j$ for $j \in \mathcal{Q}_b^{(k)}$ after receiving $G, X$ and before the training of $\mathcal{A}^{(k)}$ ($\mathcal{A}$ does not have access to the labels anyway). So Eq. (13) holds for the random drawings of all $y$'s. Now divide both sides of Eq. (13) by $n$ and use Eq. (11), we have

$$
\frac{1}{n} l_{NC}(\mathcal{A}^{(k)} | G, X, Y) \leq \frac{1}{n} \sum_{i=1}^{n} \sum_{c=1}^{C} \left[ \lambda(\mathcal{M}^*)_{j,c} \min_{j \in \mathcal{Q}_b^{(k)}} |(\mathcal{A}^{(k)})_{i,c} - (\mathcal{A}^{(k)})_{j,c}| + L((\mathcal{M}^*)_{i,c} - (\mathcal{M}^*)_{j,c}) \right] + \sqrt{\frac{L \log(1/\delta)(n-b)}{2n^2}}
$$
$$
\leq \sum_{i=1}^{n} \sum_{c=1}^{C} \left[ \frac{\lambda}{n}(\mathcal{M}^*)_{j,c} \min_{j \in \mathcal{Q}_b^{(k)}} |(\mathcal{A}^{(k)})_{i,c} - (\mathcal{A}^{(k)})_{j,c}| + \frac{L}{n}((\mathcal{M}^*)_{i,c} - (\mathcal{M}^*)_{j,c}) \right] + \sqrt{\frac{L \log(1/\delta)}{2n}}
$$
(14)

## F  HOEFFDING'S INEQUALITY

We attach the Hoeffding's inequality here for the completeness of our paper.

**Theorem 2** ([Hoeffding, 1963]). *Suppose $Z_1, \ldots, Z_n$ are independent random variables such that $a_i \leq Z_i \leq b_i$ almost surely for $1 \leq i \leq n$. Then we have*

$$
Pr \left[ \frac{1}{n} \sum_{i=1}^{n} Z_i - E \left[ \frac{1}{n} \sum_{i=1}^{n} Z_i \right] > t \right] \leq exp \left( -\frac{2n^2 t^2}{\sum_{i=1}^{n}(b_i - a_i)^2} \right)
$$
(15)