# OpenReview forum: "Active Learning Framework for Incomplete Networks"
_auai.org/UAI/2024/Conference — UAI 2024 poster_

### Official Review · Reviewer_5sBN · 2024-03-19

**Q2-1 Originality-Novelty:** 2
**Q2-2 Correctness-Technical Quality:** 3
**Q2-5 Clarity Of Writing:** 3

**Q1 Summary And Contributions:**

The paper proposes an active learning methodology coupled with GNN in order to tackle the node classification task. The main idea is to use the GNN to provide evidence for nodes that are important to be chosen by the active learning mechanism. The methodology works in rounds where for each round, a GNN is trained using available labels and then used to generate predictions for unlabelled nodes (and edges). This prediction is then used to determine the set of nodes for the next round of active learning. The proposal is evaluated on 4 different benchmark datasets and performance is compared to several other approaches. Results are competitive (for the parameters evaluated).

**Q2-3 Extent To Which Claims Are Supported By Evidence:**

2: Fair: the main claims are somewhat supported by evidence (but the experimental evaluation may be weak, or does not match entirely with the claims, important baselines may be missing, proofs contain important ideas but lack rigor, algorithmic details are only discussed superficially, references are imprecise, assumptions are not sufficiently motivated or explicated, etc.).

**Q2-4 Reproducibility:**

3: Good: key resources (e.g. proofs, code, data) are available and key details (e.g. proofs, experimental setup) are sufficiently well-described for competent researchers to confidently reproduce the main results.

**Q3 Main Strengths:**

+ Using active learning in the context of node classification in graphs is an important and timely topic.
+ Using GNNs for active learning is an interesting idea for a novel methodology.

**Q4 Main Weakness:**

- It is not clear how the output of the GNN is converted into the class prediction vector for a node $u$ and the edge prediction vector for neighbors of node $u$. This is a fundamental step that is not described.
- The effectiveness of the proposed methodology was not properly evaluated.
- The set of parameters used in the evaluation and comparison is very limited.

**Q5 Detailed Comments To The Authors:**

Although the methodology is interesting, there are a few concerns:
1) How is the prediction vector for the class of node $u$ generated from the embedding produced by the GNN?
2) How is the prediction vector for the neighbors of node $u$ generated from the embedding produced by the GNN?
These are too important issues that must be clearly described in the main text of the paper.
Moreover, Equation (2) should be defined on the predicted probability $\hat{Y}^{(k)}$ and not the actual class label $\tilde{Y}^{(k)}$. The same for Eq. (3), but maybe this is just a typo (not clear in Alg. 1 either).

A second main concern is that the proposed methodology has not been evaluated on itself. For example, it is important to understand if multiple rounds is indeed better than a single round, since this is fundamental for validating the proposed methodology. Moreover, what is the best number of rounds? Given a budget $B$, we can have a single round of active learning or up to $B$ rounds. Is B/2 rounds really worse than B/4 rounds? Another important consideration is the evaluation of the loss function. How important is to have a loss function that balances edges and nodes? There is a high computation cost on considering all edges, and it would be interesting to understand if this is truly needed.

Last, the evaluation scenarios have very limited range of parameters. The budget was varied from 200, 230 and 260, and there is no reason for this range. Moreover, the networks have different sizes in the sense that these budgets are not comparable. It would be much more interesting to see budgets going from 1% , 2%, 5%, 10%. 20%... 90%.

**Q9 Complying With Reviewing Instructions:**

Yes

---

> ### Author Rebuttal · Authors · 2024-04-05
>
> Dear reviewer, we are very grateful for your support and constructive feedback. Due to the length limit of the rebuttal, we answer your question concisely.
>
> > 1. How is the prediction vector for the class of node u generated from the embedding produced by the GNN?
> - As mentioned in the first sentence of section 4.3, the model output is the prediction for nodes. Specifically, the embedding generated by the GNN is passed through a linear layer followed by a softmax function. This process produces a probability distribution over the classes for each node, which constitutes the prediction vector for the class of node $u$.
>
> > 2. How is the prediction vector for the neighbors of node u generated from the embedding produced by the GNN? and concern in Equation (2).
> - The prediction vector for the neighbors of node $u$ is generated using a similar process to that of node $u$ itself as I explain above.
> \usepackage{amssymb}
> - Regarding Equation (2), I believe there might be a misunderstanding. In this context, the actual class label $\tilde{\textbf{Y}}^{(k)}$ is used in the $I\kern-0.15em P$ to denote the probability distribution over the classes, following by Hongyun Cai et al. [2017] in their paper (https://arxiv.org/pdf/1705.05085.pdf) on page 4, section 4.2.1. Thus, the use of an actual class label $\tilde{\textbf{Y}}^{(k)}$ in Equation (2) is not a typo. Similarly, for Equation (3) and Algorithm 1, it is not a typo.
>
> > A second main concern is that the proposed methodology has not been evaluated on itself and an important consideration is the evaluation of the loss function.
> - Our experiments show that multiple rounds of active learning outperform a single round because they provide more comprehensive information about nodes and edges, leading to more accurate node scores. However, there is a trade-off between the number of rounds and execution time (fewer rounds result in shorter execution times but lower model performance). The optimal number of rounds is found to be 8 through experimentation.
> - Regarding the loss function, our approach is motivated by the desire to have the model focus on nodes with many lost edges in the initial epochs, and then shift its focus to the final task of node classification in the later epochs. This necessitates a loss function that can effectively balance the importance of edges and nodes. Our experiments with combining two loss functions have shown that this approach results in better performance. This indicates that considering all edges, despite the high computational cost, is indeed beneficial for the overall effectiveness of the model.
>
>
> > Last, the evaluation scenarios have very limited range of parameters.
> - We have experimented with several budgets and chosen three specific budgets: 200, 230, and 260. Our goal is to compare how the same budget performs across datasets of different sizes. Dividing budgets according to ratios of nodes could be very insightful. However, there are 12 baselines and 4 datasets, so displaying all the results for each budget would take up too much space. Therefore, we selected three representative budget values to include in the paper. For more detailed results with different budgets, you can refer to the table below, which shows the experimental results on the Cora dataset with various budget values. Because of the rebuttal's length limitation, we present notable baselines.
>
>
> |               | 140                           | 170                           | 200                           | 230                           | 260                           | 290             |
> |---------------|-----------------              |-----------------              |-----------------              |-----------------              |-----------------              |-----------------|
> | Uncertainty   | $\underline{75.6}$ $\pm$ 0.8  | 77.3 $\pm$ 1.2                | 78.7 $\pm$ 0.7                | $\underline{80.6}$ $\pm$ 0.9  | 80.5 $\pm$ 1.2                | $\underline{81.4}$ $\pm$ 1.1  |
> | AGE           | 74.8 $\pm$ 1.3                | 76.8 $\pm$ 0.7                | 77.6 $\pm$ 0.9                | 77.7 $\pm$ 1.2                | 79.6 $\pm$ 0.2                | 80.2 $\pm$ 0.7  |
> | GraphPart     | 70.8 $\pm$ 1.4                | 71.4 $\pm$ 1.4                | 72.8 $\pm$ 1.8                | 73.7 $\pm$ 1.2                | 75.1 $\pm$ 0.6                | 76.2 $\pm$ 1.4  |
> | GraphPartFar  | 75.1 $\pm$ 0.5                | 76.3 $\pm$ 0.3                | 77.8 $\pm$ 0.3                | 78.4 $\pm$ 0.5                | 78.4 $\pm$ 0.6                | 79.5 $\pm$ 0.5  |
> | ALIN          | $\textbf{76.2}$ $\pm$ 0.4     | $\textbf{78.1}$ $\pm$ 1.5     | $\textbf{79.8}$ $\pm$ 1.3     | 80.4 $\pm$ 0.6                | $\textbf{81.6}$ $\pm$ 1.0     | $\textbf{82.2}$ $\pm$ 1.3  |
> | ALINFar       | 75.1 $\pm$ 1.1                | $\underline{77.5}$ $\pm$ 0.8  | $\underline{78.9}$ $\pm$ 1.2  | $\textbf{81.2}$ $\pm$ 1.1     | $\underline{81.4}$ $\pm$ 1.7  | 81.3 $\pm$ 1.6  |

---

### Official Review · Reviewer_VZkE · 2024-03-22

**Q2-1 Originality-Novelty:** 3
**Q2-2 Correctness-Technical Quality:** 3
**Q2-5 Clarity Of Writing:** 3

**Q1 Summary And Contributions:**

This paper addresses the problem of active learning on incomplete graphs. Existing methods either completely ignore edge information or rely on unreliable "translated features" that become ineffective when links are missing. The proposed work named ALIN overcomes these limitations by introducing a novel framework designed for use with Graph Neural Networks (GNNs) that exploits both node and edge information for effective node selection. The article outlines the key components of ALIN, including edge-based scoring, two-phase training, and a joint loss function. It provides a comprehensive examination of ALIN's effectiveness, drawing on both theoretical analysis, notably Theorem 1, and extensive experimental validation. The theoretical analysis, centered around Theorem 1, offers fundamental insights into ALIN's operational boundaries and inner mechanisms. These insights are complemented by a wide-ranging series of experiments, which rigorously test ALIN's robustness and flexibility across various graph neural network models and data types.Experimental results show that ALIN outperforms several methods from the literature (both general-purpose methods that are agnostic to the graph structure and methods tailored for graph-structured data).

**Q2-3 Extent To Which Claims Are Supported By Evidence:**

3: Good: the main claims are supported by convincing evidence (in the form of adequate experimental evaluation, proofs, (pseudo-)code, references, assumptions).

**Q2-4 Reproducibility:**

2: Fair: key resources (e.g. proofs, code, data) are unavailable but key details (e.g. proof sketches, experimental setup) are sufficiently well-described for an expert to confidently reproduce the main results.

**Q3 Main Strengths:**

- The paper proposes a novel active learning framework (ALIN) specifically designed for incomplete graphs, a domain not well addressed by existing methods. ALIN incorporates both node and edge information for query selection, which is a new approach in this context.
Soundness/Technical Quality:

- The paper provides a solid theoretical foundation for ALIN, explaining how focusing on uncertain nodes improves classification.
The experiment design is well-structured, comparing ALIN and its variants to various baselines on multiple benchmark datasets with different sizes and characteristics.

- It presents clear and compelling experimental results with tables showing the performance of each method on various datasets and budget constraints. The results consistently demonstrate that ALIN outperforms existing baselines, statistically supporting the claims of its effectiveness.

- The paper mentions using real-world benchmark datasets (Cora, Citeseer, Pubmed, Coauthor-CS), which are publicly available, facilitating reproducibility of the experiments.

**Q4 Main Weakness:**

- While the paper provides the specific code for ALIN in the supplementary material and offers a detailed description of the framework, the absence of a publicly available code repository, such as a Github link, weakens the reproducibility of the results presented in the paper.

-Lack of analysis regarding execution times of the ALIN algorithm and its variants.

**Q5 Detailed Comments To The Authors:**

- I recommend that the authors consider explicitly linking the experimental results to the implications of Theorem 1. This would help clarify the correspondence between theory and empirical observations.
- The mathematical expression L(k) in Equation (1) represents the combined loss function, which consists of both node classification loss l_{NC} and link prediction loss l_{LP}. However, the specific formulation of L(k) has not been explicitly defined. Moreover, it is not clear how the budget b (the maximum number of updated node labels) is integrated into the equation.
- In the paper, three hyperparameters are used, in particular \beta for calibrating l_{NC} and l_{LP}, \alpha as the weight parameter for the amalgamation of \phi(k)_{NS} and \phi(k)_{ES}, resulting in the composite score \phi(k)_{CS}. The supplementary material analyzes different values of \alpha and \beta, but overall, the retained values of the hyperparameters in the experiments are not clear.
- The specific budget values mentioned in the study (200, 230, and 260) require a more detailed justification regarding their selection. It would be beneficial for the clarity and transparency of the study to explain the process or criteria that led to these specific values.
- Theorem 1 should appear in Section 4.3 for clarity and improved organization.

**Q9 Complying With Reviewing Instructions:**

Yes

---

> ### Author Rebuttal · Authors · 2024-04-05
>
> Dear reviewer, we thank you for your valuable feedback and constructive questions. Due to the length limit of the rebuttal, we answer your question concisely.
>
> > The absence of a publicly available code repository, such as a Github link
> - We will provide the GitHub link after the paper is published. Due to the requirement for author anonymity and double-blind review, we have not provided the GitHub link.
>
> > Lack of analysis regarding execution times of the ALIN algorithm and its variants.
> - Calculating execution times in this context is not entirely appropriate. This is because execution times depend on several factors, such as training time. We have experimented with calculating the execution times during the query phase of the baselines and found that they are not the bottleneck (the comparison results are below). Therefore, we have not included it in our experiments and discussion. Moreover, works like AGE and GraphPart also do not measure execution times.
> We calculated the execution times (in seconds) during the query phase of the baselines 10 times on the Cora dataset and got the average.
> | Random| Density| Uncertainty | CoreSet| Degree| Pagerank| AGE| FeatProp| GraphPart| GraphPartFar| ALIN | ALINFar      |
> |---|---|----|---|---|---|----|----|----|-----|----|----|
> | 0.001002 $\pm$ 0.000001| 0.48 $\pm$ 0.06| 0.008 $\pm$ 0.0006| 0.09 $\pm$ 0.04| 0.0009 $\pm$ 0.0001| 0.015 $\pm$ 0.0007| 0.5 $\pm$ 0.04| 7.3 $\pm$ 1.3| 1.78 $\pm$ 0.05| 2.17 $\pm$ 0.14| 0.082 $\pm$ 0.007 | 0.081 $\pm$ 0.005|
>
> Experiments are conducted on the Windows computer equipped with Intel Core i5 10200H 2.40GHz CPUs, 16GB RAMs, and NVIDIA GTX 1650Ti 4GB.
>
> > explicitly linking the experimental results to the implications of Theorem 1
> -  We have only implicitly linked the experimental results to Theorem 1. In Section 4.1.2, we mentioned the efficiency of our entropy-based node scoring approach, which is crucial for the active learning framework. Theorem 1 proves that choosing the most uncertain nodes leads to low node classification loss, which is the basis of our ALIN and ALINFar methods. In Section 5.2, we observed that ALIN and ALINFar outperform baseline methods, supporting the theorem's implications.
>
>
> > The specific formulation of L(k) has not been explicitly defined and concern about the budget b.
> - The combined loss function $\mathcal{L}^{(k)}$ is defined as the sum of the node classification loss $l_{NC}$ and the link prediction loss $l_{LP}$, weighted by a hyperparameter $\beta$. This is stated before Equation (1), and the explicit formula for $\mathcal{L}^{(k)}$ can be found in line 18 of Algorithm 1 on page 12 of the paper.
> - Regarding budget $b$, which represents the maximum number of node labels that can be updated, it is used in the node selection process rather than directly in the loss function. Specifically, after computing the scores for the nodes, the top b nodes with the highest scores are chosen for querying and labeling. This selection process is described in line 11 of Algorithm 1 on page 12.
>
> > the retained values of the hyperparameters in the experiments are not clear.
> - In our paper, we used two hyperparameters, $\alpha$ and $\beta$. For $\alpha$, which balances $\phi_{NS}^{(k)}$ and $\phi_{ES}^{(k)}$ to form $\phi_{CS}^{(k)}$, we found $\alpha = 0.5$ to perform best, as detailed in Supplementary section C. For $\beta$, which calibrates $l_{NC}$ and $l_{LP}$, we initially set $\beta = 0.05$ and followed the "Step Growth" function for adjustments, based on the results in Table 5 of Supplementary section D.
>
>
> > explain the process or criteria that led to these specific values.
> - The selected budget values of 200, 230, and 260 were chosen due to space constraints in presenting results for 12 baselines and 4 datasets. These values are representative and provide a concise overview. For more detailed results across different budgets, please refer to the extended table for the Cora dataset. Notable baselines are highlighted due to the rebuttal's length limitation.
>
> || 140| 170| 200| 230| 260| 290|
> |----|------|------|------|----|-----|----|
> | Uncertainty| $\underline{75.6}$ $\pm$ 0.8| 77.3 $\pm$ 1.2| 78.7 $\pm$ 0.7| $\underline{80.6}$ $\pm$ 0.9| 80.5 $\pm$ 1.2|$\underline{81.4}$ $\pm$ 1.1|
> | AGE| 74.8 $\pm$ 1.3| 76.8 $\pm$ 0.7| 77.6 $\pm$ 0.9| 77.7 $\pm$ 1.2| 79.6 $\pm$ 0.2| 80.2 $\pm$ 0.7|
> | GraphPartFar| 75.1 $\pm$ 0.5| 76.3 $\pm$ 0.3| 77.8 $\pm$ 0.3| 78.4 $\pm$ 0.5| 78.4 $\pm$ 0.6| 79.5 $\pm$ 0.5|
> | ALIN| $\textbf{76.2}$ $\pm$ 0.4| $\textbf{78.1}$ $\pm$ 1.5| $\textbf{79.8}$ $\pm$ 1.3| 80.4 $\pm$ 0.6| $\textbf{81.6}$ $\pm$ 1.0| $\textbf{82.2}$ $\pm$ 1.3|
> | ALINFar| 75.1 $\pm$ 1.1| $\underline{77.5}$ $\pm$ 0.8| $\underline{78.9}$ $\pm$ 1.2|$\textbf{81.2}$ $\pm$ 1.1| $\underline{81.4}$ $\pm$ 1.7| 81.3 $\pm$ 1.6|
>
>
> > Theorem 1 should appear in Section 4.3 for clarity and improved organization.
> - Theorem 1 does indeed appear in Section 4.3. However, due to its length, the text of Theorem 1 appears on a different page.

---

### Official Review · Reviewer_1pgS · 2024-03-22

**Q2-1 Originality-Novelty:** 3
**Q2-2 Correctness-Technical Quality:** 2
**Q2-5 Clarity Of Writing:** 3

**Q1 Summary And Contributions:**

This paper proposed a new framework on Active Learning for Incomplete networks, namely ALIN framework. Their contribution includes following main aspect: new node scoring approach by introducing edge scores which gives more effective query node; Novel joint loss function combines both node classification and link prediction task; Their experiments get better performance, not only prediction, but also robustness compared to other existing methods.

**Q2-3 Extent To Which Claims Are Supported By Evidence:**

3: Good: the main claims are supported by convincing evidence (in the form of adequate experimental evaluation, proofs, (pseudo-)code, references, assumptions).

**Q2-4 Reproducibility:**

2: Fair: key resources (e.g. proofs, code, data) are unavailable but key details (e.g. proof sketches, experimental setup) are sufficiently well-described for an expert to confidently reproduce the main results.

**Q3 Main Strengths:**

This paper explores the advancements in active learning for incomplete networks, which I believe to be significant progress in current methods. I find most of their evidence to be persuasive, although there are some aspects that I still need to fully understand. Additionally, their experimental setup seems reasonable, and they have conducted thorough comparisons with existing methods, demonstrating commendable efforts in their research. The overall writing is pretty good and easy to follow as read through the whole paper.

**Q4 Main Weakness:**

In this paper, I have a few concerns. It appears that the paper solely compares node classification performance using the Macro-F1 metric. Considering your model also incorporates edge score loss, what about the evaluation of link prediction performance? Or will this model be able to do link prediction? Secondly, how does the author assess the robustness of the model? I didn't come across much relevant data regarding its robustness in the results.

**Q5 Detailed Comments To The Authors:**

I think this is overall a decent work and comments and questions are listed above. what about the evaluation of link prediction performance? how does the author assess the robustness of the model?

**Q9 Complying With Reviewing Instructions:**

Yes

---

> ### Author Rebuttal · Authors · 2024-04-05
>
> Dear reviewer, we thank you for your kind review. We are excited that you appreciated the contribution and the novelty of our work. We respond to your comments below.
>
> > In this paper, I have a few concerns. It appears that the paper solely compares node classification performance using the Macro-F1 metric. Considering your model also incorporates edge score loss, what about the evaluation of link prediction performance? Or will this model be able to do link prediction? Secondly, how does the author assess the robustness of the model? I didn't come across much relevant data regarding its robustness in the results.
> - The primary objective of our approach is to integrate edge score loss and edge scores to accommodate graphs with missing edges. While querying based on node information is standard practice, our intuition is that targeting nodes with a significant number of missing edges can provide additional insights for the model. Regarding link prediction, our model is indeed capable of performing this task. However, it is crucial to fine-tune the trade-off hyperparameters between node scores and edge scores, as well as between node loss and edge loss. We use link prediction as a supportive task; therefore, we have omitted its evaluation for brevity and focused primarily on the node classification task.
> - To assess the robustness of our model, we demonstrate its consistent and strong performance across various datasets and under different budget constraints, in comparison to baseline models. This robustness evaluation is an essential aspect of our research, ensuring that our model is not only effective but also adaptable to diverse scenarios.
>
> > I think this is overall a decent work and comments and questions are listed above. what about the evaluation of link prediction performance? how does the author assess the robustness of the model?
> - Thank you for your questions. We have addressed these questions above.

---

### Official Review · Reviewer_2TCD · 2024-03-23

**Q2-1 Originality-Novelty:** 2
**Q2-2 Correctness-Technical Quality:** 3
**Q2-5 Clarity Of Writing:** 2

**Q1 Summary And Contributions:**

This paper proposes an active learning method ALIN for incomplete networks, which trains a graph neural network (GNN) to generate node embedding based on the losses of node classification and link prediction. Experiments on four datasets have been conducted to verify the effectiveness of the proposed method.

**Q2-3 Extent To Which Claims Are Supported By Evidence:**

2: Fair: the main claims are somewhat supported by evidence (but the experimental evaluation may be weak, or does not match entirely with the claims, important baselines may be missing, proofs contain important ideas but lack rigor, algorithmic details are only discussed superficially, references are imprecise, assumptions are not sufficiently motivated or explicated, etc.).

**Q2-4 Reproducibility:**

2: Fair: key resources (e.g. proofs, code, data) are unavailable but key details (e.g. proof sketches, experimental setup) are sufficiently well-described for an expert to confidently reproduce the main results.

**Q3 Main Strengths:**

S1. Since GNNs are heavily dependent on the quality of the input graph and real-world graph are often incomplete, the active learning problem on incomplete graphs is interesting.

S2. The query strategy of combining node and edge scores to select the most informative node seems reasonable and the corresponding theoretical analysis is provided in Appendix.

S3. The paper is well organized and easy to follow.

**Q4 Main Weakness:**

W1. The node score (Eq. 2) is based on the entropy of the prediction of GNN, which seems to be a slight extension of uncertainty.

W2. The experimental results in Table 3 show that the improvements of the proposed methods are not significant, especially for the method ALINFar.

W3. The experimental settings are not clearly described, e.g., how to generate the incomplete graphs? Random removing some edges?

**Q5 Detailed Comments To The Authors:**

Q1. According to the results in Table 3, the comparison method AGE, which combines centrality, density and uncertainty for query strategy, performs better than ALINFar in datasets Pubmed and Coauthor-CS. Does this means that more metrics should be adopted into the node score?

Q2. Since ALIN performs better than ALINFar on most of datasets, ALINFar is not necessary in this paper.

Q3. Some typos should be fixed, e.g., there are five missing edges in Fig. 1 (k=0), phi_ES^{k} in the para. 3 of page 5.

**Q9 Complying With Reviewing Instructions:**

Yes

---

> ### Author Rebuttal · Authors · 2024-04-05
>
> Dear reviewer, we sincerely thank you for your encouraging words and valuable, constructive feedback. We answer the questions below. Please let us know if you have any follow-up questions.
>
> > W3. The experimental settings are not clearly described, e.g., how to generate the incomplete graphs? Random removing some edges?
> - In section 5.1, titled "Experimental Setup," we describe the process of generating incomplete graphs. To create these graphs, we remove 30% of the total edges from the complete graphs. Specifically, at the preprocessing stage, we remove edges uniformly at random from the edge list of each graph. Additionally, all baselines run on the same resulting incomplete graph.
>
> > Q1. According to the results in Table 3, the comparison method AGE, which combines centrality, density and uncertainty for query strategy, performs better than ALINFar in datasets Pubmed and Coauthor-CS. Does this means that more metrics should be adopted into the node score?
> -  While the results in Table 3 show that AGE outperforms ALINFar on the Pubmed and Coauthor-CS datasets, this does not necessarily imply that incorporating more metrics into the node score is beneficial. ALINFar is closely related to ALIN, with the primary difference being the initial node selection process, as discussed in Section 4.1.1. In our experiments, as detailed in Section 5.2, ALIN demonstrates superior effectiveness on larger datasets like Pubmed and Coauthor-CS with smaller annotation budgets, typically ranging from 200 to 230. However, when the budget increases to around 260, the performance of the baselines becomes quite similar. Therefore, the effectiveness of the methods is not significantly different with a small variance. This marginal improvement can be attributed to performance saturation at higher budget levels, rather than the inherent advantage of incorporating additional metrics into the node score.
>
> > Q2. Since ALIN performs better than ALINFar on most of datasets, ALINFar is not necessary in this paper.
> - For ALINFar, we aim to explore more case studies in the InitNodes phase beyond just randomly choosing initial nodes. Additionally, ALINFar demonstrates good results in Table 3, specifically on the Citeseer dataset.
>
>
> > Q3. Some typos should be fixed, e.g., there are five missing edges in Fig. 1 (k=0), phi_ES^{k} in the para. 3 of page 5.
>
>
> >  there are five missing edges in Fig. 1 (k=0)
> - I believe there might be a misunderstanding regarding Fig. 1 ($k=0$). Initially, there are indeed five missing links in the graph. However, after querying nodes and updating edges, three of these links are restored. When we forward this information to the GNN, we make predictions about nodes and links. The graph shown in the last part of $k=0$ is the predicted graph, which may contain incorrect link predictions. Therefore, it will differ from the initial graph, where five links were missing.
>
>
> > phi_ES^{k} in the para. 3 of page 5.
> - Thank you for your comment. This was a typographical error on our part.

---

### Meta-Review · Area_Chair_fx8J · 2024-04-21

I found the active learning framework for GNN very interesting. Even though there authors can provide better baselines (in addition to F1 scores) and more applications (in addition to link prediction and classification), they have provided enough evidence that their method works.